# Exploring Community Roles in Managing Childhood Illnesses in Vhembe District, Limpopo: Perspectives from Nurses and Caregivers

**DOI:** 10.3390/ijerph22111757

**Published:** 2025-11-20

**Authors:** Livhuwani Tshivhase, Idah Moyo

**Affiliations:** Department of Health Studies, College of Human Sciences, University of South Africa, Pretoria 0003, South Africa; idahbandamoyo@gmail.com

**Keywords:** child and paediatric health, community health services, caregivers, health care, nurses, stakeholder engagement

## Abstract

Despite notable progress in reducing childhood morbidity and mortality, achieving Sustainable Development Goal 3 remains a challenge in sub-Saharan Africa, where many children under five die before accessing formal healthcare services. This study explored the roles of the community in the implementation of the Integrated Management of Childhood Illness (IMCI) programme from the perspectives of caregivers and professional nurses. Using an interpretative phenomenological analysis design, 18 participants were purposively selected from four primary healthcare facilities. Data was collected through audio-recorded interviews, transcribed verbatim and analysed using the IPA framework. The findings underscore the critical role of community health workers (CHWs) within the Integrated Management of Childhood Illness (IMCI) framework, particularly in health promotion, child assessments and follow-up home visits. Support from early childhood development educators and community leaders further enhances these efforts. Nurses highlighted mobile health teams as vital for delivering integrated services, though challenges such as limited transport and inadequate training hinder CHWs’ effectiveness. Community-based care offers a cost-effective, accessible model in low-resource settings by leveraging local structures. Strengthening the connection between communities and formal health systems is essential. To sustain IMCI, investment in CHW-led initiatives, including training and logistical support, is recommended to improve service delivery and child health outcomes.

## 1. Introduction

To achieve Sustainable Development Goal (SDG) 3, ensuring healthy lives and promoting well-being for all at all ages, it is envisaged that globally, neonatal mortality should drop from 27.8 million to 22.7 million by 2030 [1]. UNICEF’s global study established that sub-Saharan Africa had the highest neonatal mortality rate in 2018, at 28 deaths per 1000 live births [2]. The same study reported that sub-Saharan Africa had the second highest infant mortality (58 deaths per 1000 live births) after West and Central Africa (73 deaths per 1000 live births) [2].

Available evidence on current global, regional and local trends of child morbidity and mortality rates demonstrates that despite the significant progress in containing childhood morbidity and mortality, the attainment of SDG 3 remains elusive [1,3,4]. This calls for crafting, developing and implementing innovative and proven strategies such as Community Integrated Management of Childhood Illness (C-IMCI) that will reverse the trends while improving child survival and the quality of life of children under five at the community level.

According to World Health organization, the three components of Integrated Management of Childhood Illness (IMCI) and interventions all encompass both curative and disease preventative as well as health promotive activities [5]. The IMCI strategy aims to improve child survival and development through three main pillars: (i) training health workers on improved diagnosis and treatment measures; (ii) strengthening health systems for child health services; and (iii) providing community and household interventions that address predisposing factors to childhood illnesses [5,6].

The focus of the programme is on enhancing case management skills, strengthening health systems and improving community health practice [5]. In addition, Winch and LeBan indicated that improving the quality of care provided at health facilities alone was not effective in reducing childhood morbidity and mortality [7]. In that context, the crucial role played by C-IMCI cannot be overemphasised. Implementing activities that will improve family and community practices and empowerment of mothers, caregivers and other community health workers on aspects such as exclusive breastfeeding, immunisations and home hygiene is critical, hence the emphasis on this third pillar [8]. In addition, the use of C-IMCI becomes pivotal, particularly when addressing the importance of programme sustainability and the need for local ownership [9]. Due to gaps in the healthcare delivery system such as shortage of both human and material resources in low-resource settings, the involvement of families and communities in childcare has emerged as an important stopgap measure for improving the quality of life for the children under five [10,11].

The IMCI comprises preventative and curative components that are implemented through collaborative efforts among families, communities and health facilities [12]. The World Health Organization (WHO) indicates that the community component encompasses key family practices which assist the family and community in appropriate care-seeking, education about nutrition, home case management and adherence to recommended treatment and community involvement in health services planning and monitoring [13]. Evidence from a study conducted in Bangladesh demonstrates that integrating community-based and health facility interventions significantly enhances the utilization of IMCI services [14]. C-IMCI can be delivered through outreach by facility-based health providers or implemented at the community level using community health workers [15]. Community health workers (CHWs) play a pivotal role in delivering primary healthcare services, particularly in low-resource settings. Serving as a vital link between communities and primary healthcare facilities, CHWs provide health education, promote disease prevention and offer basic care at the household level [16]. Their proximity to and trusted relationships with community members enable them to identify health issues early and facilitate timely interventions. Importantly, CHWs contribute significantly to enhancing health literacy by translating complex health information into accessible knowledge, empowering individuals and families to make informed health decisions. This function is especially critical in the context of under-five child health, where improved health literacy can lead to better prevention, early care-seeking behaviour and adherence to treatment regimens. Understanding and strengthening the health literacy role of CHWs is therefore essential for building resilient health systems and advancing progress toward universal health coverage.

Effective community involvement in the IMCI strategy requires the formation of collaborative partnerships and the sharing of responsibilities among caregivers, community health workers and healthcare professionals [16]. This collaborative approach is essential for addressing child health challenges and achieving improved health outcomes. To ensure the delivery of quality childcare services within the community, caregivers must actively monitor the physical health, emotional well-being and behavioural patterns of children, and respond promptly to emerging health needs.

Moreover, the capacity of caregivers plays a pivotal role in the successful development and care of children. As such, healthcare professionals are tasked with empowering caregivers through targeted coaching and the provision of relevant health information [17]. Evidence from existing literature underscores the importance of strengthening community-based initiatives aimed at enhancing child healthcare practices. Therefore, this study explored the roles of the community in the implementation of the IMCI programme from the perspectives of caregivers and professional nurses.

## 2. Materials and Methods

The interpretative phenomenological analysis (IPA) design was utilised to gain insight into the experiences of nurses and caregivers of children under five with reference to the community contribution to the IMCI programme. The nurses who participated in the study were professional nurses who worked in child health clinics whereas the caregivers were either parents and or grandmothers of children under five accessing services in those facilities. Through this design, researchers have the best opportunity to understand and explore the innermost deliberation of study participants’ “lived experiences” [18]. This approach enabled the study participants to narrate their stories based on their lived experiences [19,20]. Through this design, the researchers were able to explore and ask crucial questions on aspects that were narrated by the study participants to gain an in-depth understanding of their experiences [20].

The IPA approach has a three-fold focus: interpretive, double hermeneutic and idiographic in nature [18]. The purpose is to examine the lived experiences of the individual by drawing from the concepts of phenomenology, hermeneutics and ideography [21]. This approach has been used in other studies and proved to be useful in exploring experiences in healthcare settings [18]. The phenomenological aspect of IPA is that it focuses on how individuals narrate their experiences and perceive them [18]. These three aspects enable the researcher to make sense of and understand the phenomenon under study and consider the unique experience of each participant not comparing them with all other participants’ experiences [20]. In this manner, rich descriptions of individual cases are arrived at. Therefore, this enabled the researchers to have an in-depth understanding of nurses and caregivers’ experiences and perception of the role played by communities in the IMCI activities.

The study was conducted in four of the eight primary healthcare centres of Vhembe district, in rural Limpopo. Communities are largely dependent on public healthcare due to low medical insurance coverage, high unemployment and poverty levels. Linguistic diversity (mainly Tshivenda and Xitsonga), female-headed households and cross-border migration in Musina further shape healthcare demand and service delivery across these centers. Selected for the study were four high-volume healthcare facilities that provide services for children under five years. These facilities provide healthcare services for children under five that include immunisation, growth monitoring and the treatment of minor ailments, among others. Eighteen participants were purposively and conveniently selected for the study. These comprised nurses and caregivers of children under five accessing services at the four primary healthcare centres. Both categories of study participants had the relevant experiences about the contribution of the community in the IMCI programme and were available at the study sites during data collection.

Data collection commenced on 23 March 2022 and 25 May 2022. Data was collected by the first author who had worked for 20 years in similar setting and understood the cultural dynamics in the area and the research assistant who was a master’s student in nursing. As a nurse researcher with prior experience in community health, the first author was aware that her background could influence how she interpreted participants’ narratives. She therefore maintained a reflective journal throughout the study to bracket her assumptions and remain open to the lived experiences shared by the participants. This took place in a private consultation room in the primary healthcare facility secured for that purpose. During the data collection period, participants (caregivers) were recruited as they queued at the clinic to access services. The participants who agreed to participate were requested to sign informed written and verbal consent forms prior to each interview. Arrangements were made with the nurses to ensure that these participants did not lose their spot in the queue. Some interviews happened prior to consultations depending on the numbers in the queues while others were interviewed after consultations. The nurses were interviewed at times convenient for them (for example during their lunch times). The researcher established rapport through greeting of potential participants in their local languages (Tshivenda, Sepedi, Xitsonga) and introducing themselves. The researchers ensured that participants comprehended all essential aspects of the study including its purpose and data collection procedures before signing the consent forms.

The researcher requested cell phone numbers from the participants to facilitate sharing research findings for member-checking purposes. To ensure the privacy and comfort of the participants, the researcher pasted a “Do not disturb” sign on the door of the interview room. The researcher suspended her preconceptions during the data collection exercise (bracketing) to enable participants to express their concerns and make claims on their own terms [20]. Participants were selected based on specific inclusion criteria. For caregivers, eligibility required being either a parent or guardian with a minimum of two years’ experience in caring for a child within the study setting. For professional nurses, inclusion was limited to those who had received formal training in the Integrated Management of Childhood Illness (IMCI) strategy and had at least three years of experience in delivering healthcare services to children under the age of five. The professional nurse who coordinates and supervises community health workers was purposefully recruited for the study. Data were collected using a semi-structured interview guide. A semi-structured interview guide was employed in accordance with IPA methodology, which prioritizes the exploration of participants lived experiences through flexible yet focused dialogue. The grand tour question was: What are the roles of the community in the implementation of the IMCI programme? Probes that were based mostly on participants’ responses were used to ascertain clarity. This approach is widely endorsed in IPA literature for its ability to facilitate rich, detailed accounts and support the interpretive process central to phenomenological inquiry [22,23].

The interview guide was piloted on four participants working in two of the facilities who met the inclusion criteria for the study, but the findings are not included in the study. Pilot study is intended to inform and improve the main study, not contribute to its final outcomes. Including such data may compromise methodological consistency and the credibility of the study [24,25]. The aim of the pilot study was to ascertain whether the questions in the interview guide were clear to participants before the actual study was embarked upon [18]. English was used for collecting data from study participants that were nurses, while Tshivenda was used for most of the caregivers who preferred it. A voice recorder was used together with field notes to collect data after an explanation was carried out and the participants consented both verbally and by signing the consent forms. Data saturation was reached at participant number six for nurses, but the researcher continued up to participant number eight. Regarding caregivers, data saturation was reached at participant number eight, but two additional participants were interviewed to confirm such. Data saturation is the point at which data become redundant, repetitive or no new information emerges while collecting data from participants [25]. A total of eighteen participants participated in the study as presented in Table 1a,b. The interviews lasted for 45 min to an hour with each participant.

Interpretative Phenomenological Analysis (IPA) framework was employed to analyse the data, following established methodological guidelines [20]. The audio-recorded interview data were transcribed verbatim into written text. Interviews conducted in Tshivenda were initially translated and transcribed into English by the first author, who is fluent in both Tshivenda and English and possesses a deep understanding of the cultural context and practices relevant to the study setting. Two researchers analysed the transcripts independently using the IPA framework. The researchers are experienced mature qualitative researchers with over 30 peer reviewed publications. The open coding of each transcript was conducted by an independent coder, who is a full professor, an expert in qualitative research and is well published. As part of the data analysis process, the researchers followed the structured steps of Interpretative Phenomenological Analysis (IPA) to ensure methodological rigor and depth of interpretation. Each researcher independently engaged with the data by reading and rereading the transcripts and listening to the audio recordings multiple times to immerse themselves in the participants lived experiences. The steps followed were (1) reading and rereading the transcript; (2) notetaking and developing emergent themes; (3) clustering the emergent themes; (4) crafting a master table of themes (this was composed of superordinate themes, subthemes and extracts from the interviews); (5) examining and comparing the similarities between the master tables of the themes; and (6) compiling a single master list of themes and subthemes. Thereafter, the researchers met the independent coder to compare and discuss their respective master tables of themes. Consensus was reached through collaborative deliberation, guided by criteria such as thematic relevance, recurrence across transcripts, coherence within the thematic framework and richness of supporting data. The first author shared field notes, a reflective journal with the independent coder and the second author to support the themes that were shared. Non-verbal cues such as tones, pauses and gestures were also shared to support the interpretation drawn from participants. The final master table (Table 2) reflects this consensus and represents a comprehensive synthesis of the CHWs’ roles as interpreted from the qualitative data [19,20].

Protecting the rights of the institution is an important principle of research [26,27]. To ensure that this principle was adhered to, the researcher sought and obtained ethical approval from the Sefako Makgatho University Research Ethics Committee (SMUREC/H/334/2021:IR). Permission to conduct the study was also obtained from the Limpopo Department of Health. After obtaining information about the objectives and background information related to the study and before each interview was conducted, participants signed an informed consent form. As part of the consent process, participants were also told that participation was purely voluntary, and that consent could be withdrawn at any time without prejudice. In addition, study participants also consented to audio recordings of the interviews and publication of the study findings. Participants were assured that data would be handled confidentially and that the results would be reported in a way that ensured anonymity. Anonymity and confidentiality were maintained by removing all identifiers and using pseudonyms. The transcribed data were only reviewed by the researchers. To enhance confidentiality, data were stored securely on a password-protected computer. The study participants were provided with a detailed information sheet that explained the details of the study. After reading and understanding all the information in the detailed information sheet that explained the details of the study, the participants signed an informed written consent form.

Measures were taken to enhance trustworthiness and rigour [28]. To ensure trustworthiness, the research should satisfy four criteria: credibility, transferability, dependability and confirmability [29]. Credibility refers to the accuracy of the findings [26]. To enhance credibility, all interviews were audio recorded and transcribed verbatim. Member checking, peer evaluation and co-coding were used to enhance credibility. An independent coder, who was not involved in data collection, reviewed and independently conducted the analysis to ensure rigour and credibility [27]. Transferability relates to the ability of the findings to be transferred to other contexts. A “thick description” of the research context was provided to allow the reader to assess the transferability to other situations. The researcher thoroughly described the research context, participants and data collection method. To enhance dependability, the research process is described in detail to enable another researcher to replicate the study [29]. To ensure an audit trail, a step-by-step description of methods was carried out. All records were kept on a password-secured computer. Confirmability was enhanced using bracketing to minimise researcher bias. To enhance authenticity, verbatim extracts from the interviews were utilised and member checking was conducted: The researchers re-engaged the participants through phone calls and went over the individual transcripts with each participant a week after the interviews. The verbatim transcripts were summarised and then discussed with each participant in a simplified and accessible manner to ensure clarity and mutual understanding [30,31]. This facilitated the checking for accuracy of information contained in the transcript and the participants confirmed that their perspectives were adequately captured and represented.

## 3. Results

Participants for the study comprised eight professional nurses and ten caregivers of children under five years who were accessing child healthcare from four primary healthcare centres in Vhembe district. Demographic data of the participants are presented in Table 1a,b below.

Four themes emerged from data analysis, namely the supportive role of community healthcare workers, community outreach for child health services, partnering with the community stakeholders in IMCI implementation and challenges facing community healthcare workers. Table 2 presents the four themes and eleven subthemes developed from the study findings.

### 3.1. Supportive Role of Community Health Workers

Caregivers in this study stated the supportive role they are afforded by the community healthcare workers such as providing community awareness of child health services, assessment of sick children and conducting follow-up and home visits.

#### 3.1.1. Community Awareness of Child Health Services

Caregivers reported the extensive support they received from the ward-based outreach teams. Nurses also alluded to the fact that community healthcare workers were playing a pivotal role in creating community awareness of childcare. The following excerpts illustrate that:


*“They take an active role by creating awareness about childhood illness on existing challenges such as diarrhoea.”*
(Conny, 33 years, nurse)


*“The community health-care workers usually gather us in our village and teach us how to take care of our children.”*
(Langanani, 32 years, guardian)


*“Home-based caregivers also play a pivotal role in demand generation for community health activities. They also participate in immunisation awareness campaigns.”*
(Alice, 39 years, nurse)

#### 3.1.2. Assessment of the Sick Child

Caregivers indicated that they were empowered through health education to the extent that they were able to assess their children’s condition whether they could manage them at home or had to refer them to the healthcare facility. The following extract demonstrates that:


*“I have been taught by nurses and homebased carers on how to check my children. I usually bring the children to the clinic for them to get medical treatment quickly. If the child gets a fever at night, I give them water to drink and I use a wet towel to use on their face, if I have Panado in the house then I give them Panado. I make sure the children always wash their hands, I don’t allow them to pick up food that fell on the floor; when it’s cold I make sure they are wearing warm clothes; I also don’t allow the children to play with water to avoid flu; if the child got hurt and [is] bleeding, I treat the wound by cleaning it up and stopping the bleeding.”*
(Mumsy, 28 years, guardian)

#### 3.1.3. Follow-Up and Home Visits

Caregivers indicated that community healthcare workers followed up with them at their homes. The Road to Health card was used to profile each household. Children missing immunisations were referred to healthcare facilities for vaccination as is evidenced by the excerpts below.


*“Community health workers usually come to count us at home, check our health. They also ask for the child’s Road to Health card to check if the child had all the injections and refer us if children had missed doses of immunisation.”*
(Phindulo, 29 years, guardian)


*“In the community, we sometimes get support from the community workers who visit us in our homes, give health education on childcare. They also assist us to assess if the child needs to be sent to the health-care facility or [can be] observed at home.”*
(Maria, 37 years, guardian)


*“We send the homebased carers, because nowadays we have health-care workers who work outside. We send them to go and make follow-up or call the mother. They conduct follow-up visits through telephone or physical home visits.”*
(Brenda, 42 years, nurse)

### 3.2. Community Outreach for Child Health Services

Nurses were able to report on the community outreach for child health services that are being conducted in those rural villages that are far from the clinics. Mobile teams comprising different categories of nurses were reported to be providing integrated child healthcare services, community follow-up of children with delayed milestones and growth monitoring for children under five years.

#### 3.2.1. Provision of Integrated Child Healthcare Services

Nurses indicated that they provide disease-prevention activities for children under five years such as immunisations, health education on proper hand washing before feeding children and preparing food, and on prevention and management of child injuries. They had the following to say:


*“Communities far from the facility are usually serviced by mobile teams who go out and render immunisations for under-five children, treat minor ailments on those children and give the mothers health education on a variety of things. They educate mothers on hand washing prior to child feeding and food preparation.”*
(Timothy, 47 years, nurse)


*“As nurses we usually go out to the community and health educate caregivers on proper feeding of children, proper waste disposal and prevention and management of child injuries at home.”*
(Ivy, 30 years, nurse)

#### 3.2.2. Community Follow-Up of Children with Delayed Milestones

Nurses reported that for those children who were assessed and given follow-up dates, they do follow up through conducting home visits to find out if the problem had been resolved or not. Nurses also follow up on children they have seen who fail to thrive or have delayed milestones, in order to refer them timeously. They had this to say:


*“All children we assess and find they need follow-up are treated and given a follow-up date. If they fail to come to facilities for follow-up, we follow them to their homes to check on them.”*
(Lilly, 41 years, nurse)


*“We do follow-up of children under five years who had fever or with delayed milestone to see how the child is doing. We also could send the community health-care workers to follow up all those with problems.”*
(Alice, 39 years, nurse)

#### 3.2.3. Growth Monitoring for Children

Nurses indicated that among the activities performed in community outreach was the weighing of children under five to monitor their growth. The participants had this to say:


*“We usually weigh children to see if the child is gaining or losing weight. We also do assess the feeding of the child, asking the mother whether the child is still on breastfeeding or what is she feeding the child?”*
(Elelwani, 53 years, nurse)


*“We check the Road to Health card to see if each child is on track. We check if the child is below the 3rd centile or above. Every visit the child is weighed, and we compare with the previous weight.”*
(Andani, 47 years, nurse)


*“I make sure I bring the child to the clinic for her weekly immunisation and weighing. At home I make sure she is being breastfed a lot.”*
(Maria, 39 years, guardian)

### 3.3. Partnering with Community Stakeholders in Delivering Child Healthcare Services

#### 3.3.1. The Role of Community Leaders in Awareness

Caregivers indicated the support the community leaders were offering through community meetings. Community leaders created slots for health awareness and also emphasised the importance of attending such fora. The following excerpts demonstrate that:


*“We are often called to the chief’s kraal and those trained health-care workers, and a clinic nurse join to address health issues in the presence of the chief. They announce all awareness meetings and immunisation campaigns there and the chief or community leaders chairing meetings will emphasise the importance of bringing children to be immunised.”*
(Londani, 56 years, guardian)


*“Almost all monthly meetings in the community hall, there will be a nurse giving a health talk either on treating diarrhoea in children or the importance of monthly family check-up.”*
(Emmah, 67 years, guardian)

#### 3.3.2. The Role of Preschools in Facilitating Continuity of Childcare

Nurses indicated that preschool teachers were taught how to support children who were on treatment and about measures that could prevent children from becoming ill. The extracts below show that.


*“So, you find that because the parents will be shifting the blame associated with diarrhoeal disease to the preschools, information on proper hygiene, food handling and preparation practices is given to preschool workers. Some children are given their medication by preschool teachers. A collaborative role does exist between health-care facilities and preschools, for example, the dietician from the Department of Health does regular support visits to the preschools.”*
(Elelwani, 53 years, nurse)


*“Children spend a lot of time in the preschools. Since the preschools deal with under-fives, it is important for the preschool workers to be informed about the IMCI strategy. When children fall sick, the parents and or caregivers are called to collect the sick child to take the child to the health facility. In most cases the preschool teacher does counselling that influences the guardian to take the child to the health facility.”*
(Brenda, 42 years, nurse)

### 3.4. Challenges Facing CHWs

Nurses further indicated the barriers experienced by community health workers that hinders effective and sufficient community health services. Community health workers were reported to be facing challenges of transport to travel around all households, non-acceptance in some families and inadequate training related to community child health services.

#### 3.4.1. Transport Challenges Faced by CHWs

Nurses have reported that CHWs frequently face difficulties reaching certain households due to long distances and lack of transportation. These challenges hinder their ability to conduct home visits and complete child assessments. Reported experiences include:

*“Most of the children who present at our clinic with severe complications such as dehydration in this facility come from villages that are far” Some households are too far to reach on foot, making it impossible to conduct proper profiling and assessment of children in those areas. This challenge of distance has been expressed by community health care workers*.(Conny, 33 years, nurse) 


*“There have been instances where CHWs were requested to do follow-up of children who missed doses of immunization or missed scheduled clinic visits, however, they failed because of long distance. Although CHWs serve as our extended hands for home visits, the distances involved are often too great for them to manage without transport.”*
(Alice, 39 years, nurse)

#### 3.4.2. Limited Acceptance of CHWs in Certain Households

Some CHWs have reported challenges in gaining access to certain homes during their outreach activities. This lack of acceptance has hindered their ability to collect important health information, particularly concerning children. The following excerpts are in support:


*“As the contact person responsible for the coordination of CHWs activities in the clinic, I have witnessed situations where they submitted the weekly report with missing data. The reason in the reports for missing elements will be denied access to the household.”*
(Brenda, 42 years old, nurse)


*“I remember last year when I was doing home visit with the CHWs and I was walking behind them, I saw them being locked out by an elderly caregiver even when children were present and playing in the yard. They only allowed them in when they recognised me through nurses’ uniform and allowed us inside. They were only allowing me to see the newborn saying they prefer a nurse to see the newborn only. So, without my presence they were going to leave without assessing the children.”*
(Timothy, 47 years old, nurse)

#### 3.4.3. Inadequate Training of CHWs

Nurses have expressed concern that while CHWs play a vital role in identifying and referring sick children, many lack sufficient training to provide basic care effectively. This gap in knowledge and skills affects the quality of child assessments and follow-up care. Their reported experiences include:


*“I have observed that some of the CHWs are unsure of how to measure and interpret mid-upper arm circumference, as they were never formally trained and are simply learning from peers.”*
(Andani, 47 years old, nurse)


*“Since my allocation to work with CHWs three years ago, I do not remember them going for refresher courses. “Several CHWs mentioned that their last training on assessing children and chronic patients was over 10 years ago. There is a clear need for refresher workshops to update their skills in child assessment.”*
(Lilly, 41 years old, nurse)

## 4. Discussion

The purpose of the study was to explore the role of the community in the implementation of the integrated management of childhood illness from the perspectives of nurses and caregivers of children under five years. The findings highlight the critical supportive role played by CHWs in promoting child health through community-based awareness initiatives. These awareness efforts serve to educate caregivers on distinguishing between child health conditions that can be managed at home and those requiring referral to healthcare facilities. Community awareness is essential for increasing public knowledge of available health programmes and services [32]. CHWs were reported to conduct home visits during which they assess children who appear unwell and provide counselling to caregivers on appropriate actions by either home-based care or referral to a health facility. Furthermore, CHWs or nurses are responsible for follow-up of children who have previously visited healthcare facilities and are scheduled for return visits. During these follow-ups, CHWs remind caregivers to bring the child back to the clinic, thereby ensuring continuity of care and adherence to treatment plans.

Caregivers in this study benefited from community-based awareness initiatives, where both nurses and CHWs collaborated to disseminate knowledge and promote practices that support child health. The contributions of CHWs were reported as essential and advantageous to families and communities. Home visits by community health workers have been associated with improved infant feeding practices among mothers, and a reduced likelihood of children experiencing wasting or symptoms of common childhood illnesses within the subsequent two weeks [33]. Additionally, community health workers play a critical role in improving access to preventive care services during the early childhood period [34]. Moreover, CHWs are acknowledged as integral components of community health systems, undertaking diverse responsibilities such as health education, advisory support, rehabilitation and facilitation of group-based interventions [35]. Community engagement plays a vital role in strengthening health outcomes, with social mobilization serving as a key strategy to enhance participation and responsiveness [36]. Community health workers contribute significantly by empowering households and addressing underlying determinants of child health, nutrition and development [37]. Furthermore, well-structured health education programmes at the community level can improve caregivers’ ability to recognize symptoms of common childhood illnesses and identify danger signs that require urgent clinical attention [38]. Without any interventions, children under five are at risk of morbidity and mortality. Participants reported that outreach teams deliver integrated child health services to communities located at a distance from healthcare facilities. These services include growth monitoring, follow-up care for children discharged with scheduled return dates, immunization, treatment and referrals when necessary. Outreach activities conducted by mobile community teams are essential in reaching populations with limited access to healthcare facilities. These activities are in line with the functions of community healthcare workers as outlined in the training manual on caring for children in the community [39]. The findings of this study demonstrate that the use of a comprehensive and integrated management approach is pivotal in improving child health outcomes [40,41]. Integrating disease prevention, diagnosis and treatment of minor childhood ailments while accounting for the social, cultural and economic context of the communities has been shown to enhance the effectiveness of child health interventions [42]. CHWs are closer to and available when the health facilities are closed to ensure appropriate home-based treatment for children [43,44]. They are involved in conducting home visits to promote maternal and child health nutrition, monitor growth, do home assessments, capacitate the families and make referrals when needed [45,46]. Studies have shown that educating and empowering families enhances their ability to implement appropriate home care practices, including maintaining adequate fluid intake, ensuring proper nutrition and preventing infection through hygienic behaviors [47].

Community leaders and Early Childhood Development (ECD) centres educators were found to be supporting childhood illness management in the communities through awareness on child healthcare practices and continuity of care at the ECD centres. Community leaders are expected to supervise, monitor and support the selected community health workers when they start working in the local community [45]. Involvement of stakeholders in the health of children could therefore improve community child healthcare practices.

While the role of the CHWs in the childhood illness management is proving to be critical, it is important to develop strategies that enhance the quality of services implemented in the community.

On the other hand, nurses outlined the difficulties the CHWs faces whilst executing such noble activities that promote child health and prevent childhood diseases. CHWs are expected to travel on foot through villages to carry out their duties, even when the homes they need to visit are located up to 10 km away from healthcare facilities or their own residences. Although CHWs are regarded by both the nursing staff and the caregivers as essential contributors to the promotion of child health within communities, there remains a critical need to equip them with foundational clinical skills. Adequate training is necessary to ensure that the care they provide meets acceptable standards of quality and effectiveness [48]. Equipping community health workers with comprehensive training, ongoing mentorship and regular refresher courses is essential for ensuring the effective delivery of child health services [49]. The support is crucial, particularly in under-resourced healthcare systems like South Africa’s, where CHWs serve as a foundational component of primary healthcare delivery. Another challenge experienced by the CHWs was the denial of access to some household even though they were to assist the families in the management of childhood illness. Similar findings are supported by authors who indicated that CHWs face critical challenges of poor resources, conflicting roles for remuneration issues and inadequate training to execute their duties [50]. This suggests that, despite their commitment to improving community health outcomes, CHWs continue to face significant challenges that hinder their effective participation in child healthcare services.

This study findings reveal the importance of health literacy in shaping caregivers’ understanding and management of childhood illnesses. This aligns well with the global health priorities outlined in the Sustainable Development Goals (SDG 3) aimed at ensuring healthy lives and promoting the well-being of all ages. The study’s emphasis is on the CHWs roles that promote access to health information and services for caregivers. Through the awareness-raising efforts of community health workers (CHWs), health literacy within communities is enhanced, thereby enabling caregivers to prevent disease, seek timely medical care and adhere to treatment regimens which plays a critical effort in reducing child morbidity and mortality.

Emphasizing the distinct education and training of professionals and caregivers in child and pediatric health is essential, as it serves as a foundational strategy for effective health promotion and the prevention of disease.

The study was confined to selected primary healthcare facilities within a single district, which may limit the generalisability of the findings to other districts. However, the study shared valuable insights into the roles of CHWs in managing childhood illnesses. Although a few studies have explored this topic specifically in Vhembe, similar research in districts such as KwaZulu-Natal and Ekurhuleni have highlighted both the challenges and contributions of CHWs in child health service delivery [48,51].

## 5. Implications

For practice, strengthening CHW support systems can significantly contribute to improved child health outcomes and the successful implementation of the IMCI strategy. Improved logistical and transport support is essential to ensure CHWs can reach remote households and deliver equitable services. Enhancing collaboration between CHWs, nurses, community leaders and preschool staff can promote a more coordinated approach to child health and ensure continuity of care. Increasing community engagement and sensitisation could improve household acceptance of CHWs and foster trust in community-based health initiatives.

For policy makers, the findings support the need for policy reinforcement of CHWs’ roles within the IMCI and primary healthcare frameworks. Policymakers should consider integrating CHWs into the health system with clear role definitions, supervision structures and adequate remuneration.

## 6. Conclusions

This study highlighted the multifaceted roles of CHWs in supporting the implementation of the IMCI programme within rural communities. CHWs were found to be instrumental in raising community awareness about child health, conducting assessments of sick children and performing follow-up and home visits. Their involvement in outreach services such as immunisation campaigns, growth monitoring and follow-up of children with delayed milestones demonstrates their critical contribution to bridging the gap between health facilities and populations served. Furthermore, collaboration with community stakeholders, including preschools and traditional leaders, was shown to enhance continuity of care and promote community ownership of child health services and initiatives.

Despite their vital role, CHWs face significant challenges that hinder the effectiveness of IMCI implementation. These include logistical barriers such as transport limitations, inadequate training and a lack of cooperation from certain households. Addressing these challenges through capacity-building, improved resource allocation and strengthened community engagement is essential to ensure the sustainable delivery of child health services. The findings emphasise the importance of integrating CHWs into the primary healthcare system and enhancing partnerships with local structures to improve child health outcomes in rural settings.

## Figures and Tables

**Table 1 ijerph-22-01757-t001:** (**a**) Demographic data of professional nurses. (**b**) Demographic data of caregivers for children under five years.

(**a**)						
**Number**	**Participant’s Pseudo Name**	**Age in Years**	**Gender**	**Years in Practice as a Professional Nurse**		**Years Implementing IMCI**
1	Alice	39	Female	12		10
2	Brenda	42	Female	14		14
3	Andani	47	Female	20		15
4	Conny	33	Female	9		9
5	Elelwani	53	Female	20		13
6	Ivy	30	Female	7		7
7	Timothy	47	Male	18		15
8	Lilly	41	Female	15		15
(**b**)						
**Number**	**Pseudo name**	**Gender**	**Age (years)**	**Relationship with the child**	**Age of child (months)**	**Number of children under five in the household**
1	Phindulo	F	29	Mother	9 months	2
2	Maria	F	39	Mother	12 months	5
3	Lindiwe	F	22	Mother	48 months	2
4	Mumsy	F	28	Mother	36 months	2
5	Emmah	F	67	Grandmother	24 months	3
6	Langanani	F	32	Mother	36 months	3
7	Londani	F	56	Grandmother	6 months	1
8	Maria	F	37	Mother	3 months	2
9	Julia	F	34	Mother	36 months	2
10	Ndivhuwo	F	28	Mother	36 months	1

Table b: Author’s own creation.

**Table 2 ijerph-22-01757-t002:** Themes and subthemes for the study.

Theme	Subthemes
2.1 Supportive role of community healthcare workers	2.1.1 Community awareness of childcare
	2.1.2 Assessment of the sick child
	2.1.3 Follow-up and home visits
2.2 Community outreach for child health services	2.2.1 Provision of integrated child health services
	2.2.2 Community follow-up of children with delayed milestones
	2.2.3 Growth monitoring for children
2.3 Partnering with the community stakeholders in IMCI implementation	2.3.1 Role of preschools for continuity of care
	2.3.2 Role of community leaders
2.4 Challenges facing community healthcare workers	2.4.1 Transport challenges faced by CHWs
	2.4.2 Limited acceptance of CHWs in certain households
	2.4.3 Inadequate training of CHWs

Author’s own creation.

## Data Availability

The raw data supporting the conclusions of this article will be made available by the authors on request.

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
