# Peer review of "Exploring Community Roles in Managing Childhood Illnesses in Vhembe District, Limpopo: Perspectives from Nurses and Caregivers"

_ijerph, 2025, doi:10.3390/ijerph22111757_

Round 1

Reviewer 1 Report

Comments and Suggestions for Authors

Keywords

Only “caregivers” and “nurses” belong to mesh or decs terms; the rest of the terms used are neither mesh nor decs terms, and the authors should update them.

Introduction

Reference is made to sustainable development 3, but it is necessary to state the full name of this objective, not just the number to which it refers.

Information on community health workers should be provided, as they are a key part of the study developed by the authors. Therefore, the introduction should include information describing their roles and importance in the community.

Objective

It is stated that the contributions of the community to the implementation of the program will be explored, but the study uses nurses and caregivers as a sample, so the objective should focus on this area, explaining the contributions of the community from the professional community perspective of nurses and the community perspective of caregivers.

Method

The academic background of the researchers conducting the qualitative analysis of the study should be outlined, including their professional qualifications, years of experience, age, gender, and whether they live in the study area or outside it.

The sociodemographic characteristics of the four selected primary healthcare centers should also be outlined.

The characteristics of the research who established rapport should be specified and described.

The inclusion criteria should be explained in greater detail.

An explanation should be provided as to why the results obtained in the guided interview were not included in the study (lines 142-143).

The consensus criteria used by the researchers to arrive at the final master table of themes and subthemes should also be explained.

When referring to the use of an independent coder who was not involved in data collection, the characteristics of this coder should be described in the study, including age, gender, academic background, etc.

Results

Two tables of themes and subthemes should be created: one table of themes and subthemes identified in the interview with nurses and another table of themes and subthemes identified in the interview with the caregiving population.

Discussion

A paragraph should be included referring to the limitations encountered in conducting the study and whether there are similar studies with which to compare the current study.

Conclusion

A conclusion should be presented that reflects in depth on the main results, as it is brief and does not refer to any aspects of the results obtained in the research.

Reviewer 2 Report

Comments and Suggestions for Authors

The article addresses a topic within the scope of Health Promotion in a territory with very specific characteristics, revealing the persistent efforts of health professionals alongside caregivers with the aim of improving health and preventing disease. I am very grateful for reading your article, which reveals the symmetries that exist in the world; I am very grateful for your study, but also for your work!!!

I offer some suggestions for improvement in my review. As follows:

  • The abstract should clearly present the objective of the study.
  • Keywords. Change childhood diseases to Child and Paediatric Health.
  • The introduction should present the main concepts through the evidence currently available. This will allow for a better understanding of the context and description of the study.

In Materials and Methods, the third paragraph should elaborate on the type of study (interpretative phenomenological) within the qualitative research paradigm. In addition to what this approach allows, the beginning of this chapter should mention, at a conceptual level, what interpretative phenomenology is. This justifies its choice for the study's objective over other types and approaches to research.

  • The research question should be presented, taking into account the phenomenon under study and the questions that the results of this study aim to answer.
  • In addition to the objective, the phenomenon under study must be clearly stated.
  • Present the criteria for inclusion and exclusion of participants.
  • The type of participant selection must be presented, in addition to a description of the phases/stages of how it occurred. Characterise the semi-structured interview used in data collection. Is this type of interview the best choice and in accordance with phenomenology? Justify.

Given that this is a phenomenological study, I felt the lack of phenomenological writing that characterises this type of research. In which the author is guided, in writing, by the words spoken by the participants. In which non-verbal communication, the environment, and logbook/travel notes also count. In which the interview is not questions and answers, but an encounter... between two people, in which one tries to access the lived world of the other...

  • When analysing the data, I believe it is essential to present the data analysis method. The phases described seem insufficient to me. They seem to refer to Bardin's content analysis...
  • The interviews with the carers were translated by professionals qualified to do so...???

This question may seem secondary, but it is not. In phenomenological studies, it is through words and listening to what the words tell us through the voices of the participants that researchers can describe/interpret the phenomena under study. Therefore, this process must be described clearly and in detail.

  • Were the interviews returned to the participants after transcription, and were the translations also returned to the carers?

In the Conclusions, it is important to highlight the distinct Education/Training of professionals and carers in the field of Child and Paediatric Health as a determining tool for Health Promotion and Disease Prevention. The introduction of the concept of Health Literacy is relevant to the discussion and conclusions of the study. In the final part of the article, it is relevant to relate the results to what was presented at the beginning of the article, i.e., the Sustainable Development Goals established by the United Nations. It may also indirectly influence policies related to the continued reduction of the infant mortality rate and the prevalence of diseases in paediatric age in the territory where the study took place.

Therefore, authors should present the implications for the field of research, but also the implications that the results obtained, which involved professionals and carers, should have in the development of public health policies implemented by the country's authorities. For example, the difficulties that professionals face in travelling around the country to make home visits, which are essential for the continuity of health promotion and monitoring of the care provided by carers and families.

After all, research also serves this purpose, i.e., adding knowledge to the discipline allows it to be transformative for and in people's lives. This is, in fact, the true implication of research.

Round 2

Reviewer 2 Report

Comments and Suggestions for Authors

Dear authors,

I would like to express my gratitude for the changes that have been incorporated into the most recent version that was sent to me.
It is evident that the alterations you have made have considerably enhanced the quality of your work, thereby optimising the entire process and yielding pertinent conclusions and implications for future research.
I would like to express my gratitude and commend you on your commendable efforts.